# Microstructure Development and Properties of the Two-Component Melt-Spun Ni_55_Fe_20_Cu_5_P_10_B_10_ Alloy at Elevated Temperatures

**DOI:** 10.3390/ma14071741

**Published:** 2021-04-01

**Authors:** Krzysztof Ziewiec, Mirosława Wojciechowska, Irena Jankowska-Sumara, Aneta Ziewiec, Sławomir Kąc

**Affiliations:** 1Institute of Technology, Pedagogical University of Krakow, Ul. Podchorążych 2, 30-084 Kraków, Poland; miroslawa.wojciechowska@up.krakow.pl; 2Institute of Physics, Pedagogical University of Krakow, Ul. Podchorążych 2, 30-084 Kraków, Poland; irena.jankowska-sumara@up.krakow.pl; 3Faculty of Metals Engineering and Industrial Computer Science, AGH University of Science and Technology, Al. A. Mickiewicza 30, 30-059 Kraków, Poland; aziewiec@agh.edu.pl (A.Z.); slawomir.kac@agh.edu.pl (S.K.)

**Keywords:** metallic glasses, thermal properties, mechanical properties, microstructure characterization

## Abstract

The aim of this work was to investigate the features of microstructure, phase composition, mechanical properties, and thermal stability of the two-component melt-spun Ni55Fe20Cu5P10B10 alloy. The development of the microstructure after heating to elevated temperatures was studied using scanning electron microscope and in situ high temperature X-ray diffraction. The high-temperature behavior of the two-component melt-spun Ni55Fe20Cu5P10B10 alloy and Ni40Fe40B20, Ni70Cu10P20, and Ni55Fe20Cu5P10B10 alloys melt-spun from single-chamber crucible was investigated using differential scanning calorymetry at different heating rates and by dynamic mechanical thermal analysis. The results show that band-like microstructure of the composite alloy is stable even at 800 K, although coarsening of bands forming the microstructure of the ribbons is observed above 550 K. Plastic deformation is observed in the composite previously heated to temperatures of 600–650 K. The properties of the composite alloy are generally different than the properties obtained for the melt-spun alloy of the same average nominal composition produced traditionally. Additionally, the mechanical and the thermal properties in this composite are inherited from the amorphous state of alloys that are precursors for two-component melt spinning (TCMS) processing.

## 1. Introduction

Due to the continuous interest in new technologies and devices, there is a growing demand for new materials, both structural and functional, with unique characteristics related to magnetic and electrical mechanical properties. Moreover, it is advantageous if such materials have the highest possible thermal stability as well as the alloying and manufacturing costs are minimized. For this reason, the use of low-cost substrates and manufacturing technologies is attractive.

One of the ways to achieve the goal of controlling the microstructure of the element in order to obtain the appropriate performance characteristics is to produce a composite composed of materials with different properties. There are reports on the production of amorphous-amorphous and amorphous-crystalline materials [1,2,3,4,5,6] and works on controlling the phase composition and properties of these materials [7,8,9,10]. Among the methods of producing amorphous composites, technologies based on the production of powders and their amorphization are also used [11,12,13,14]. They require precursor alloys with liquid supercooled range T_g_–T_x_. However, this is possible for specific chemical compositions. An interesting idea for producing glassy composites based on using the immiscibility in the liquid state [15,16], but this approach is limited to alloys with specific chemical compositions ensuring at the same time the presence of liquid immiscibility and ability of vitrification, however achieving these goals is often contradictory. 

The two-component melt spinning two-component melt spinning (TCMS) [17,18] method does not have the above-mentioned limitations. The TCMS method potentially enables the production of amorphous materials with unique structural characteristics, obtained without the need to use high purity precursors. TCMS materials are a relatively new proposition of composite materials that requires the study of phase transformations and their influence on properties as a function of temperature. This work presents research linking the development of the properties and the phase transformations in the TCMS composites. Changes in phase composition and properties are associated with processes with different activation energies. The current research is an attempt to establish the relationships between the phase transformations, the mechanisms that control them, and the properties of the produced composite.

## 2. Materials and Methods

The following alloys were prepared in arc melting furnace: Ni_40_Fe_40_B_20_, Ni_70_Cu_10_P_20_, Ni_55_Fe_20_Cu_5_P_10_B_10_ starting from 99.95 wt% Ni, 99.95 wt% Fe, 99.95 wt% Cu, Ni-P, Cu-P, and Ni-B, Fe-B master alloys. The precursors were re-melted under an argon gettered protective atmosphere. Then, the alloys were melt-spun using helium as a protective gas. Parameters of the melt spinning process were as follows: Ejection pressure—150 kPa, linear velocity of the copper roller—40 m/s, and the crucible orifice diameter—1.2 mm. On the base of these alloys, four ribbons were produced on the melt-spinner. The first one was produced by two-component melt-spinning (TCMS), i.e., in this case, the ejection of the Ni_40_Fe_40_B_20_ and Ni_70_Cu_10_P_20_ alloys from the double-chamber crucible was preceded by heating and melting them, while the two melts were separated by a partition barrier (Figure 1). 

The other three ribbons i.e., those produced from Ni_40_Fe_40_B_20_, Ni_70_Cu_10_P_20_, and Ni_55_Fe_20_Cu_5_P_10_B_10_ alloys, were made using ejection of the melt from a traditional single-chamber crucible. TCMS ribbons samples were heated in a DSC apparatus to different temperatures i.e.,: 550, 600, 633, 650, 715, 750, and 800 K at a heating rate of 20 K/min. The cross-sections of the ribbons were polished mechanically and etched with a pure nitric acid. The metallographic specimens intended for testing on the nanoindenter were not etched. Then, the morphology and chemical composition of the cross-section of these TCMS heat treated samples as well as the samples in as-melt-spun state were analyzed by JEOL 6610 scanning electron microscope (Tokyo, Japan) with an Oxford X-ray micro-analyzer (High Wycombe, UK). The observations were made in the secondary electron image mode under acceleration voltage 20 kV and working distance 10 mm. The samples were also tested using CSM Instruments nanoscratch Berkovich micro-hardness tester (Needham, MA, USA). The parameters of the measurement were as follows—maximum load: 50 mN, loading and unloading rate: 100 mN/min. 

X-ray in situ studies of melt-spun ribbons as a function of temperature were carried out using a Siemens D5005 (Bruker-AXS, Karlsruhe, Germany) diffractometer equipped with a Cu lamp with a 30 mA and 40 kV power supply. The radiation was monochromatized with a graphite monochromator, X-ray radiation wavelength was λ_CuKα_ = 1.5406 Å and its energy 8.0478 keV. As-cast ribbons were stuck to a glass plate. The measurements were done at room temperature and at elevated temperatures, i.e.,: 553, 603, 623, 643, 663, 713, 753, 803 K, in an XRK-900 reaction chamber (Anton Paar, Graz, Austria). Diffraction patterns were registered in the range of 30–60° (2θ) using θ–θ scan with 0.04°/1 s speed. The diffraction peaks were identified using PDF 4+ database files [19].

The differential scanning calorimetry of the melt-spun ribbons were made under argon atmosphere, using NETZSCH DSC200 F3 MAIA instrument (Selb, Germany), for the following heating rates: 10, 20, 40, 60, and 80 K/min. Activation energy values were determined on the base of peak temperatures using Kissinger method. All of the four ribbons were investigated by means of dynamic mechanical thermal analyzer (DMTA) NETZSCH-DMA 242 E Artemis (Selb, Germany) to determine thermal and mechanical characteristics including storage modulus. The DMA measurements were carried out in the tension mode. The other parameters applied are the following: heating rate—2 K/min, proportional force factor—1.1, maximum dynamic force—5 N, additional static force—0 N, maximum deformation amplitude—4 ± 1 μm, frequency—1 Hz, sample length—12.1 to 12.7 mm, sample width—2.92 to 3.14 mm, sample thickness—28 ± 2 μm. The sample chamber was purged with excessive N_2_ (2500 mL/min) in order to avoid oxidation.

## 3. Results and Discussion

Figure 2 presents SEM microstructure of the TCMS alloy and EDS line scan of the samples in as-melt-spun state and as-heated state at different temperatures. Bright band-like areas are rich in Ni, Cu, and P while darker bands are enriched in Fe and depleted in Ni, Cu, and P. 

Figure 3 shows the evolution of the band microstructure in the TCMS alloy from the initial state after melt-spinning to the state after heating to different temperatures: 550, 600, 633, 650, 715, 750, and 800 K. The microstructure of TCMS alloy in the melt-spinning condition and samples heated to 550 and 600 K, is similar. The average thickness of the bands is about 0.63 ± 0.45 µm, and the bands with the maximum thickness do not exceed 2 µm. The average thickness of the bands in the samples heated to 633 and 650 K is similar to the variants from lower temperatures and amounts to 0.65 ± 0.47 µm and 0.69 ± 0.50 µm, respectively. However, in these two variants, a loss of contrast between the bands is observed. Areas of such bands are marked with arrows. For samples heated to the temperatures of 715, 750, and 800 K, a further coarsening of bands is observed, and such coarsened areas are much wider than those at lower temperatures and reach a thickness of even several µm.

In situ X-ray diffraction patterns for Ni-Fe-Cu-P-B (TCMS) samples are shown in Figure 3. Figure 4 presents in situ X-ray diffractions for Ni-Fe-Cu-P-B (TCMS) alloy. The XRD made for the TCMS alloy in as-melt-spun state and for 553 K has a wide diffraction maximum for the *2*θ angle in the range of 40° to 50°. There are no crystalline phase peaks in these two diffractograms, so it can be assumed that this alloy remains amorphous up to the temperature of 553 K. At XRD for 603, 623, 643, and 663 K, there are peaks from crystalline phases such as fcc solid solution of nickel (space group: Fm-3m, a: 3.5238 Å, PDF 00-004-0850), the tetragonal Ni_3_P compound (space group: I4, a: 8.956 Å, c: 4.388 Å, PDF 04-005-5615), and the phase isomorphic with CuNi compound (space group: Fm-3m, a: 3.5636 (5) Å, PDF 01-071-7832). Analysis of the X-ray diffraction patterns for higher temperatures i.e., 713, 753, and 803 K allows identification of similar phases to those occurring at lower temperatures as well as the following additional phases: FeNi phase (space group: Fm-3m, a: 3.575 Å, 01-071-8322), the rhombic (orthorhombic) compound Fe_1.5_Ni_1.5_B (a: 5.34 Å, b: 6.637 Å, c: 4.437 Å, PDF 04-001-9092), and FeNi_3_ compound (space group: Fm-3m, a: 3.5556Å, PDF 01-071-8324).

Therefore, as presented in Table 1, the crystallization of the samples begins with the formation of the compounds containing nickel, copper, and phosphorus. Further, at temperatures as high as 713 K, the phases containing iron and boron are formed. The present observations confirm the reports presented in earlier works [20,21], which suggest that Ni-Cu-P alloys [20] crystallize at a lower temperature than Fe-Ni-B alloys [21].

The comparison of the currently conducted SEM microstructure observations (Figure 1 and Figure 2) of TCMS tapes and in situ X-ray diffraction tests (Figure 3) shows that the coarsening of the bands occurs with greater intensity in the same temperature range in which the amorphous Ni-Fe-B component of the composite crystallizes on the matrix.

Figure 5 shows results of nanoindentation tests including hardness number, Young modulus, and appearance of the indentations. Figure 6 presents dependence of hardness number and elasticity modulus as a function of temperature. For samples in as-melt-spun condition and heated to 550 and 600 K, the hardness number is on the level of 711–784 HV. Within this range of heating temperature, Young modulus rises slightly from 140 GPa (RT) to 142 GPa for 550 K and at higher temperatures, its values remain stable and they are 132, 130, and 132 GPa, for 600, 633, and 650 K, respectively. 

For the above-mentioned temperatures, SEM micrographs of indentations (Figure 5) show the presence of multiple slip-steps markings in the vicinity of the indentation. For such conditions, where the existence of slip-steps markings is observed, one can expect that the metallic glass is ductile and the material is not prone to catastrophic cracking, without prior plastic deformation [22,23]. For samples heated to higher temperatures, both, hardness number and Young modulus have growing trends, leading finally to hardness as high as 1109 HV and elasticity modulus 163 GPa. It is worth to mention that no multiple slip-steps near the indentation are observed, for samples heated to 715, 750, and 800 K. The traces of plastic flow around the indentations for samples heated to 600, 633, and 650 K formed the pile-ups that are visible as multiple stepped arcs. This may suggest the formation of multiple shear bands. For the remaining variants, the pile up next to the trace of indenter is only observed in the form of single stepped arcs.

DSC curves obtained at different heating rates for the TCMS alloy as well as for the A, B, and C alloys melt-spun after ejection from a single-chamber crucible are shown in Figure 6. The values of temperatures for crystallization peaks derived from Figure 7 are presented in Table 2. Comparison of the crystallization onsets for the three alloys melt-spun from single-chamber crucible (Figure 7) shows that the B alloy is the most stable since the highest crystallization onset T_xB_ (687–716 K)—on the other hand, A alloy has the lowest range of crystallization onset i.e.,: T_xA_ (574–600 K). Crystallization onset T_xC_ for the C alloy is in the range of 619–648 K. These onset temperatures are intermediate between those for alloy A and B. The TCMS alloy despite the same nominal composition as C alloy, has the crystallization onset temperature closest to the corresponding value for alloy A. Possible explanation for this is that some of the structural constituents of the TCMS composite alloy have similar thermal properties as the alloy A, thus the crystallization onset temperature is close to the value measured for A alloy. 

The TCMS alloy presents the most complex crystallization sequence of all of the tested alloys (Figure 7). Some of the peak temperatures i.e.,: T_1A_ and T_2A_, for the TCMS alloy, coincide with the corresponding values for the A alloy (Figure 7, Table 2), there are also peaks positions similar to those found in B alloy i.e.,: T_1B_ and some peak positions can be refered to the peaks corresponding to the alloy C i.e.,: T_2C_. It should be noted that for the DSC curves of the TCMS variant, T_2C_ peaks temperatures are varying from 728 to 750 K. These values are closer to those found in C alloy (T_C2_) than A alloy (T_A3_). It can be explained that in A alloy, the third heat effect is very weak and barely noticeable on the DSC, while for the C alloy, the T_C2_ effect is the strongest thermal effect. In the TCMS alloy, the occurrence of places with a chemical composition similar to C alloy is limited to the boundary intermediate zones between alloy A and B alloy. Generally, it can be assumed that the TCMS alloy, DSC trace is much complex than the curves found for any of the A, B, and C alloys. On the other hand, the composite thermograms seem to inherit the peak positions from the above-mentioned alloys melt-spun from the single-chamber crucible.

Figure 8 presents Kissinger diagrams derived from the DSC plots for all of the alloys, and the activation energies are given in Table 3. It is observed that the activation energies of the reactions occurring in the TCMS alloy are lower than in A and B alloys melt-spun from a single-chamber crucible. Thus, E_2A_ for the TCMS variant is 224 kJ/mol and for alloy A, it is 264 kJ/mol. The activation energy E_1B_ for the TCMS variant is 365 kJ/mol and for alloy B is 406 kJ/mol. On the other hand, the activation energy value (E_2C_ = 415 kJ/mol) related to the transformation at T_2C_ temperature is higher for the TCMS alloy than the activation energy determined for the alloy C, which is 373 kJ/mol. 

It can be expected that the activation energies of transformations in alloys A, B, and C are different, because the liquids corresponding to the precursor liquids A and B certainly undergo some modification of the chemical composition as they pass through a nozzle. Obviously, the intermediate regions between the streams of alloys A and B, similar in composition to alloy C, will slightly differ from the composition C. 

Therefore, such difference in the values of E_2C_ activation energies observed between TCMS alloy and in C alloy, is justified. The values of the peak temperatures T_1C_ and T_2A_ of the alloys met-spun from the single-chamber crucible are similar. Therefore, taking into account the fact that TCMS alloy consists of the bands of A and B alloys as well as the compositions intermediate between compositions A and B close to composition C, it is possible that the activation energy for peaks occurring at the temperature T_2A_, is slightly lower than the activation energy value for pure alloy A melt-spun from a single-chamber crucible. The activation energy value for transformation occurring at T_2A_ i.e., E_2A_ is 264 kJ/kmol for alloy A, and the value of E1C is 192 kJ/kmol, therefore the activation energy for transformation 2A (224 kJ/kmol) in the TCMS alloy has an intermediate value between 264 kJ/kmol and 192 kJ/kmol, which may confirm the above-mentioned explanation for lowering the activation energy for the T_2A_ peak. The activation energy value of the first transformation in the A alloy is E_1A_ = 224 kJ/mol and in the TCMS alloy it is 178 kJ/mol, therefore the factors controlling this transformation in the TCMS alloy may be different than in the A alloy. This value is similar to the one reported in work [20], where 178 kJ/mol was the activation energy of the amorphous matrix and Ni_5_P_2_ formation. It is worth mentioning that the activation energy associated with Ni_3_P formation reported in work [24] was 238 kJ/mol, therefore it is possible that crystallization of the TCMS alloy has more than one stage and formation of Ni_3_P is possibly precedeed by formation of an intermediate phase. The two transformations occurring at higher temperatures, are related to higher activation energies E_1B_ and E_2C_ i.e., 365 and 415 kJ/mol, respectively. The E_1B_ value for the TCMS alloy is therefore lower than the E_1B_ value for the B alloy melt-spun from single-chamber crucible. On the other hand, the E_2C_ for TCMS alloy is higher than the E_2C_ activation energy for alloy C (i.e., 373 kJ/mol). Therefore, it is possible that the 2C transformation is controlled by factors similar to the 1B transformation in alloy B. Similarly high activation energies were observed in Fe_64_Ni_16_P_14_B_6_ and Fe_16_Ni_64_P_14_B_6_ alloys reported in work [24] i.e., 441 and 335 kJ/mol, respectively. Therefore, in our case, it is possible that the occurrence of these peaks is related to the crystallization of the compositions that contain locally more boron or more phosphorus than it is in an average composition of alloy C, which suggests that the areas of the TCMS alloy contributing to this thermal effect were not thoroughly mixed as it was the case of alloy C.

The peak temperatures for the transformations in the TCMS composite are slightly higher than those for the alloys melt-spun starting from a homogeneous liquid. This relationship is the most pronouced in the case of the T_2A_ and T_1B_, which represent the location of the strongest exothermic effects for crystallization of amorphous phases in DSC thermograms. A possible reason for increasing these temperatures in the TCMS composite is higher number of alloying elements in the transition areas between the bands formed from liquids A and B. Such chemical complexity apparently raises the crystallization temperatures of the amorphous regions. Similar observations are reported by Brechtl et al. [25], where they report that more complex chemical composition increases thermal stability. Higher temperatures of phase transformations may also be related to the increased value of mixing entropy, which also hinders diffusion. Such observations are consistent with the results obtained on high entropy alloys [26,27]. 

Figure 9 shows the temperature changes of the storage modulus for the TCMS alloy and for the alloys melt-spun from single-chamber crucibles i.e., A, B, and C alloys. In general, alloy B shows the highest values of the storage modulus, while alloy A shows the lowest values of E’. This applies to the entire temperature range of the DMTA test. When comparing the properties of the TCMS alloy to the alloy with the same nominal composition C, it is worth noting that in the entire tested temperature range, TCMS alloy has higher values of the storage modulus E’ than alloy C. Initially, with an increasing temperature, all alloy variants maintain practically the same value of the storage modulus E’. At a temperature of about 400 K for A, C, and TCMS alloys, the E’ value starts to increase. In the case of alloy B, however, a continuous and slight decrease of E’ up to the temperature of 500 K is observed. 

Of all tested alloys, only in cases of alloy A and TCMS, the changes of the storage modulus due to crystallization begin in almost at the same temperature i.e., at T_1maxA_ = 517 K and T_1maxA_ = 519 K, respectively. The first local maximum of E’ for alloy C is slightly higher, i.e., at T_1maxC_ = 548 K. The first minima for the TCMS, A and C variants occur at T_1minC_ = 621 K, T_1minA_ = 578 K, T_1minC_ = 621 K, respectively. It is worth noting that in the case of alloy A for temperatures above T_1minC_ = 621 K, the value of E’ only slightly increases up to the temperature T = 620 K. Above this temperature, E’ module increases in variants TCMS, A, and C. For these three variants of samples i.e., TCMS, A, and C, the temperatures for which the second maximum of E’ occurs are similar and are T_2maxA_ = 648 K, T_2maxC_ = 654 K, T_2maxA_ = 654 K, respectively. This may indicate that the phase transformations occurring in this temperature range similarly affect the value of the storage modulus E’. 

Among all the variants of alloys, alloy B has the highest value of the storage modulus and its stability is the highest. Thus, the first deep minimum of the storage modulus occurs at T_1minB_ = 683 K. A similar deep minimum in the E’ diagram occurs for the TCMS alloy at T_1minB_ = 684 K. It is worth noting that both alloy B and the TCMS variant undergo similar changes of the storage modulus, starting from this minimum. For these variants, the maximum value of the storage modulus occurs for a similar temperature, i.e.,: for the alloy B: T_1maxB_ = 723 K, and for the TCMS alloy: T_1maxB_ = 722 K. The alloy A after reaching the maximum value of E’ at T_2maxA_ = 654 K slightly decreases the storage modulus at the end of test, i.e., around 750 K. There is a minimum of the E’ value for alloy C at T_2minC_ = 714 K, and a maximum at T_3maxC_ = 715 K, however, it seems that this part of the plot does not have any convergence with the changes for the TCMS composite alloy. 

The observations above indicate that the plot of the storage modulus for the TCMS alloy inherits the changes of E’ occurring at elevated temperatures in amorphous alloys A and B. There are no noticeable features of the E’ diagram for the TCMS alloy, which are similar to alloy C. Therefore, it can be concluded that the influence of the transformations in the regions intermediate between bands A and B on the shape of the DSC characteristic for TCMS alloy is small. It is noteworthy that TCMS alloy has a relatively high E’ value which is larger than the storage modulus for alloy C. In comparison with alloys A and B, up to 684 K, TCMS alloy shows the lowest deviations of the E’ value with relation to the value of the storage modulus at ambient temperature E’_0_ (measured as δ = (E’_0_ − E’)/E’_0_). For TCMS alloy, the values of δ calculated for successive minima vary from −1.7% to 4.1%, while for alloy A the changes ranged from 13.3% to 43.6%. For alloy B, it was 17.1%, and for alloy C, the range was from 0.3% to 7.7%. Therefore, from all of the tested alloys up to the temperature of 654 K, the TCMS alloy is characterized by the lowest changes in the value of the E’ module. TCMS alloy, after reaching the lowest value of E’ at T_1minB_ = 684 K, rapidly increases by 23.1%, which is probably due to the content of alloy B bands in the TCMS composite, because in the alloy B, in this temperature range, there is also steep increase of E’ by 17.1%. The influence of the transformations occurring in A and C alloys in this temperature range is small due to a smooth slight decrease of the E’ value for the alloy A.

The decrease in hardness and Young’s modulus is observed for samples heated to temperatures between 550 and 650 K. Changes of properties for range of temperature between T_xA_ and T_1A_ are consistent with the effects observed in DSC thermograms in a similar temperature. Roughly, it is near the temperature where onset of crystallization of the regions rich in Ni, Cu, and P (alloy A) is observed. This amorphous phase transforms into a mixture of crystalline phases such as: FCC Ni, tetragonal Ni_3_P, and FCC CuNi. On the other hand, crystallization of amorphous phases causes increase of storage modulus E’ [28,29]. Furthermore, in this temperature range, there is a significant decrease of the storage modulus E’ in material B, which is not accompanied by crystallization in Ni-Fe-B rich regions. Therefore, it is possible that the transformations within the amorphous phase of the Ni-Fe-B regions (B alloy) are responsible for this decrease in the mechanical properties of the TCMS composite material, despite the fact that crystallization takes place in the regions enriched in Ni, Cu, and P that correspond to A alloy. Moreover, the transformations related to the crystallization of the B component (enriched in Fe, Ni, and B) result in a significant increase in hardness and Young’s modulus above 750 K. Of the storage modulus obtained by the DMTA tests, both in TCMS composite and in B alloy, also indicate a significant increase in the E’ value above the temperature of 684 K. This is in accordance with the transformations responsible for these changes in properties observed on the DSC as the peaks at T_1B_ and T_2C_.

## 4. Conclusions

(1)In the TCMS samples, two kinds of band-like areas are observed: the ones rich in Ni, Cu, and P, and the other is consisting mainly of Fe, Ni, and B. The composite crystallization takes place in two stages: the first stage consists the crystallization of the Ni-Cu-P bands, the second stage is due to the crystallization of the Fe-Ni-B bands. The coarsened band-like morphology is observed at elevated temperatures even at 800 K.(2)The beginning of coarsening is observable at the temperature of 633 and 650 K. This overlaps with the crystallization of Ni-Cu-P bands, during which Ni-FCC phases, Ni_3_P tetragonal nickel phosphide and CuNi-FCC are formed from the amorphous matrix. At higher temperatures where the crystallization of Fe-Ni-B bands occurrs and crystalline phases are formed (i.e.,: FeNi-FCC, FeNi_3_-FCC, and tetragonal Fe_1.5_Ni_1.5_B), the effect of coarsening becomes more pronouced and the thickness of chemically homogeneous bands significantly increases.(3)For the TCMS samples heated to 600, 633, and 650 K, multiple slip-steps near the indentation were observed, which suggests the formation of multiple shear bands and plastic deformation of the composite during the test after heating to this temperature range.(4)The transformations within the amorphous phase of the Ni-Fe-B regions cause temporary decrease in the mechanical properties such as HV number, Young modulus, and storage modulus of the TCMS composite in the range of temperature ca 600 and 684 K, despite the presence of the crystalline phases occurring within Cu-Ni-P regions of the TCMS alloy.(5)The increase of HV number and Young modulus in the TCMS sample heated to temperature as high as 800 K, and increase of storage modulus at T_1maxB_ = 722 K, can be due to the crystallization of amorhpous part of the composite enriched Ni, Fe, and B and formation of the FCC FeNi, FCC FeNi_3_, and Fe_1.5_Ni_1.5_B phases. This transformation can be attributed to the DSC peaks at T_1B_ and T_2C_.(6)The transformations observed in the DSC study for the TCMS alloy, are very close to these obtained for A and B alloys melt-spun from a single-chamber crucible. Moreover, there are also transformations in this composite alloy similar to those that are observed in the homogeneous melt-spun C alloy of the average nominal composition of TCMS alloy. The occurence of these transformations can be attributed to the fact that there is a small volume fraction of the TCMS alloy, which corresponds to the compositions A and B.(7)The activation energies of the transformations occurring in the TCMS alloy are lower than the corresponding values obtained for A and B alloys (i.e.,: E_1A_, E_2A_, E_1B_) melt-spun from a single-chamber crucible. Activation energy for 2 C transformation in TCMS alloy is related to the crystallization of the compositions that contain locally more boron or more phosphorus than it is in an average composition of alloy C. These observations may be due to the fact that there are areas with a chemical composition similar to the alloys A and B as well as with intermediate concentrations, similar to the homogeneous alloy C but with local concentrations of phosphorus and boron higher than in alloy C.(8)In comparison with the alloy C of the same nominal composition, TCMS alloy is characterized by a higher storage modulus in the entire studied temperature range. Similarly, TCMS alloy has a higher storage modulus than alloy A. At lower temperatures, changes in the storage modulus in TCMS alloy show similarity to alloy A, while from 684 K, changes in E’ modulus are similar to material B. Small influence of areas with averaged composition C on the course of changes of storage modulus E’ in TCMS alloy was found.(9)This work shows the unique properties of the TCMS composite in comparison with the melt-spun alloy of the same nominal chemical composition that was produced traditionally. The mechanical and the thermal properties of this composite are obviously inherited from the amorphous state of the alloys that are precursors for the two-component melt spinning.

## Figures and Tables

**Figure 1 materials-14-01741-f001:**
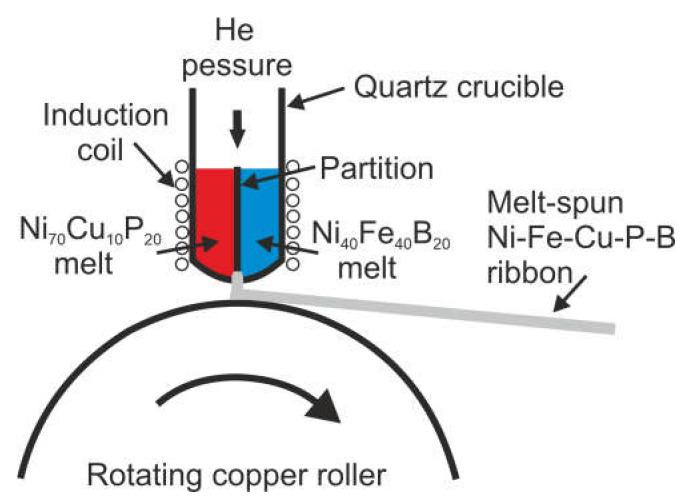
Diagram showing the method of remelting and the location of the partition separating the Ni_70_Cu_10_P_20_ and Ni_40_Fe_40_B_20_ alloys during the ejection onto a rotating copper roller.

**Figure 2 materials-14-01741-f002:**
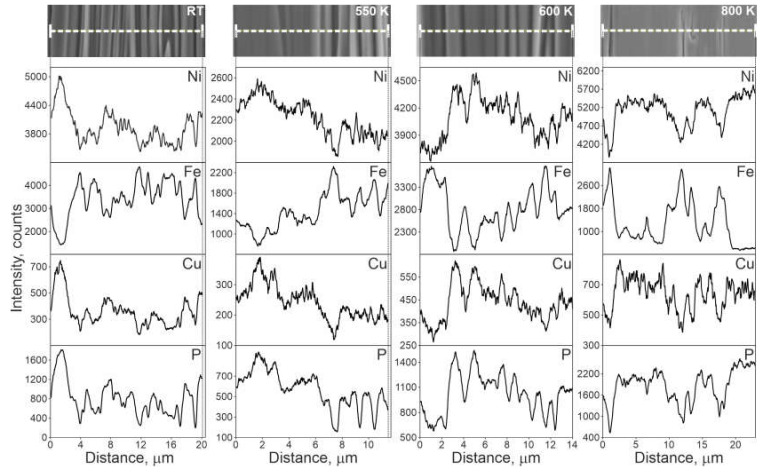
SEM microstructure of TCMS Ni_55_Fe_20_Cu_5_P_10_B_10_ alloy and EDS line scans performed on samples in as-melt-spun state and after heating to elevated temperatures at 20 K/min.

**Figure 3 materials-14-01741-f003:**
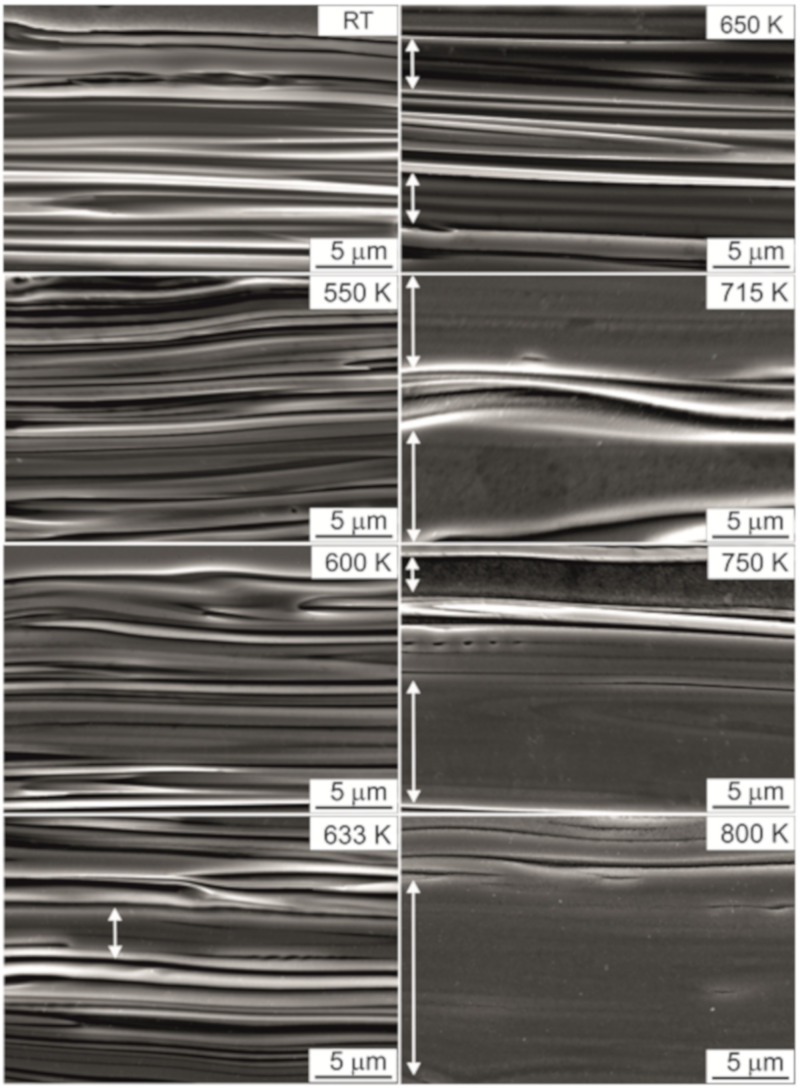
SEM microstructure of (two-component melt spinning) TCMS Ni_55_Fe_20_Cu_5_P_10_B_10_ alloy performed on samples in as-melt-spun state and after heating to elevated temperatures at 20 K/min; white arrows indicate the coarsened bands.

**Figure 4 materials-14-01741-f004:**
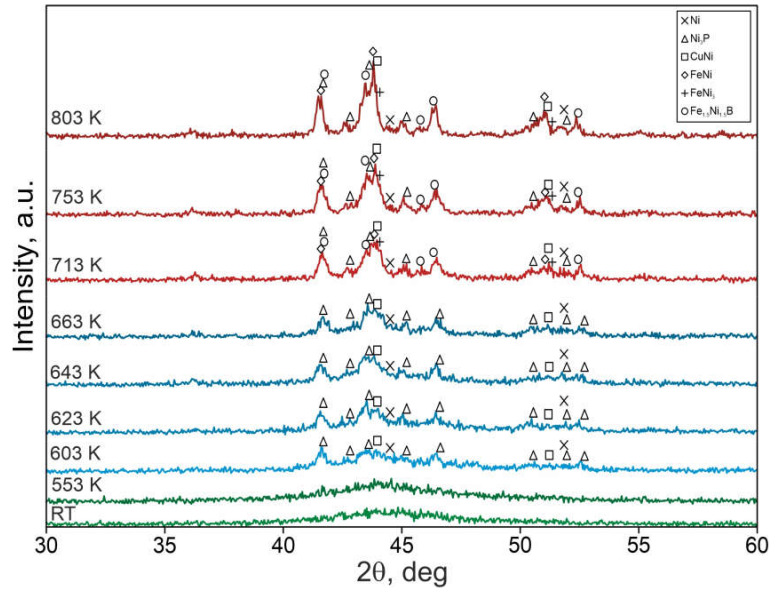
In situ X-ray diffraction patterns for the TCMS Ni_55_Fe_20_Cu_5_P_10_B_10_ alloy obtained at different temperatures.

**Figure 5 materials-14-01741-f005:**
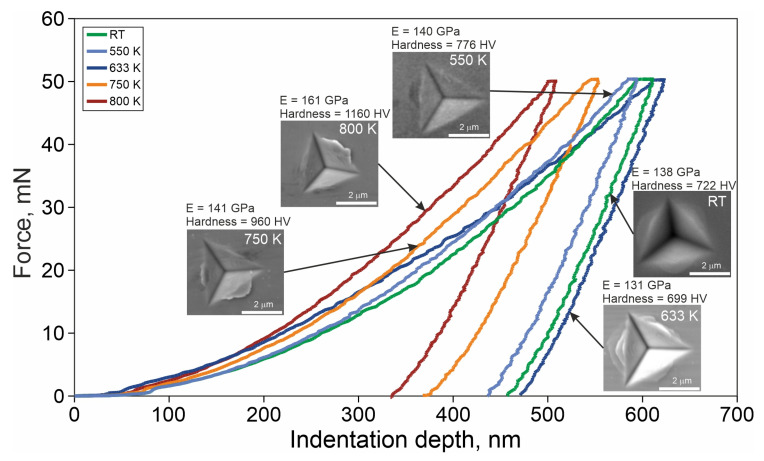
Typical nanoindentation load-depth (P-h) curves, SEM images of the indenter imprints, Young modulus, and hardness number of the TCMS Ni_55_Fe_20_Cu_5_P_10_B_10_ alloy in as-melt-spun state and after heating to different temperatures.

**Figure 6 materials-14-01741-f006:**
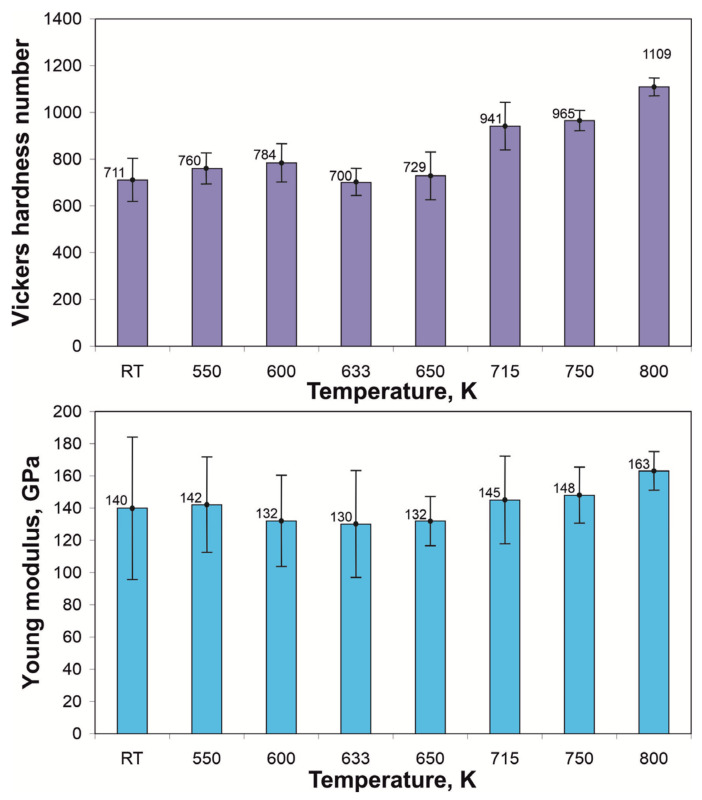
Hardness and Young modulus for TCMS Ni_55_Fe_20_Cu_5_P_10_B_10_ alloy in as-melt-spun state and samples heated to different temperatures.

**Figure 7 materials-14-01741-f007:**
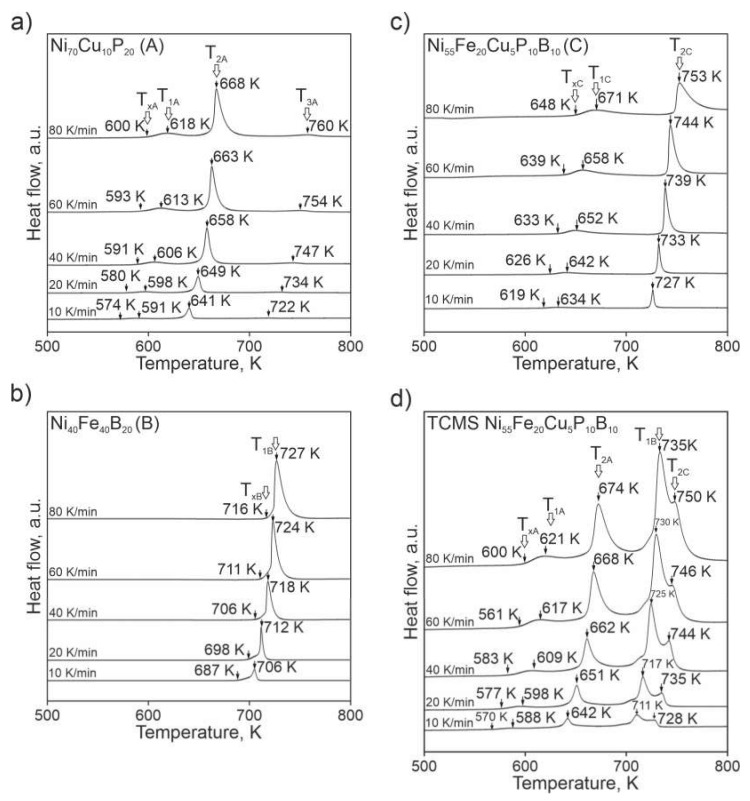
Differential scanning calorymetric (DSC) curves for different melt-spun alloys: TCMS Ni_55_Fe_20_Cu_5_P_10_B_10_, Ni_55_Fe_20_Cu_5_P_10_B_10_ (C), Ni_40_Fe_40_B_20_ (B), and Ni_70_Cu_10_P_20_ (A) performed with various heating rates; the following symbols denote: T_xA_, T_xB_, T_xC_ crystallization onset; T_1A_, T_2A_, T_3A_—temperatures of subsequent crystallization peaks for alloy A; T_1B_—crystallization peak temperature for alloy B; T_1C_, T_2C_—temperatures of subsequent crystallization peaks for alloy C; (**a**) Ni_70_Cu_10_P_20_ (A), (**b**) Ni_40_Fe_40_B_20_ (B), (**c**) Ni_55_Fe_20_Cu_5_P_10_B_10_ (C), (**d**) TCMS Ni_55_Fe_20_Cu_5_P_10_B_10._

**Figure 8 materials-14-01741-f008:**
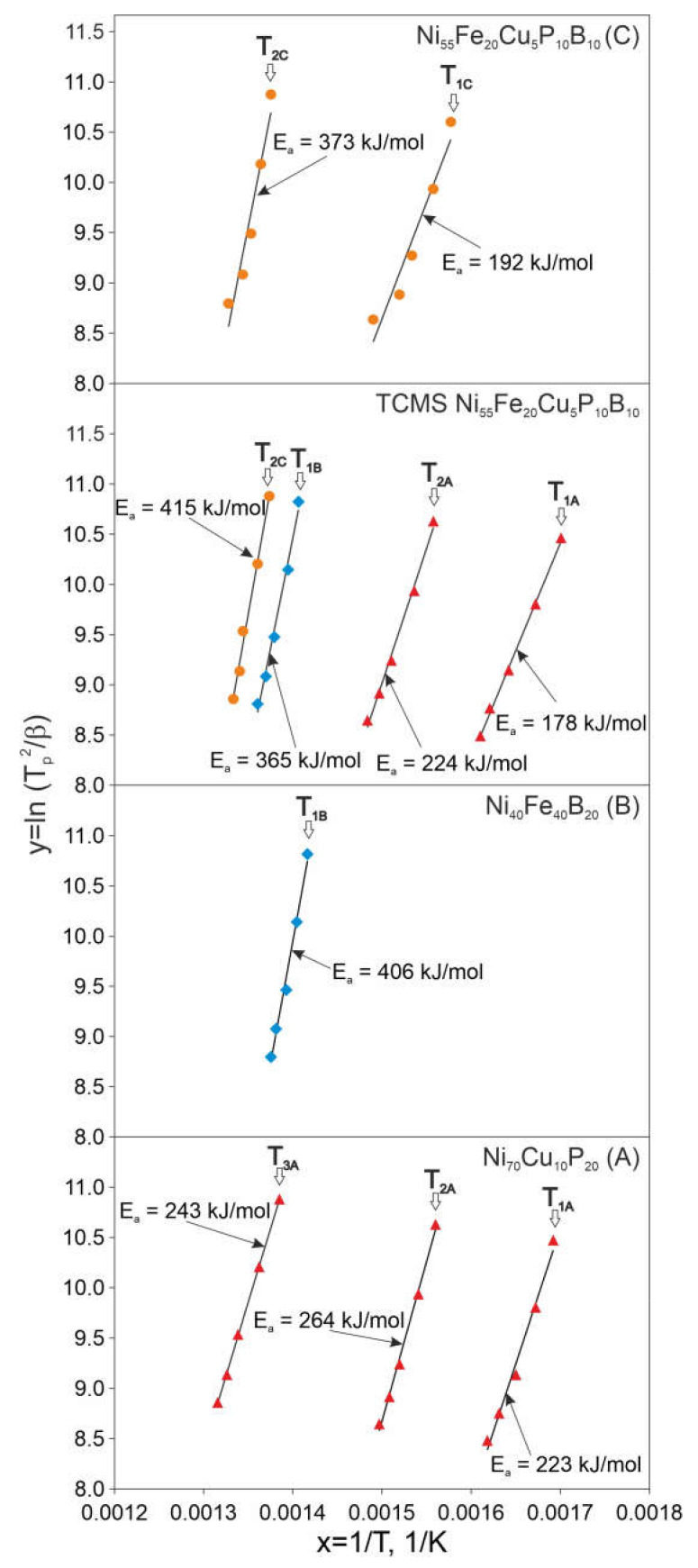
Kissinger plots constructed on the base of DSC peak temperatures for different melt-spun alloys: Ni_70_Cu_10_P_20_ (A), Ni_40_Fe_40_B_20_ (B), Ni_55_Fe_20_Cu_5_P_10_B_10_ (C), and TCMS Ni_55_Fe_20_Cu_5_P_10_B_10._

**Figure 9 materials-14-01741-f009:**
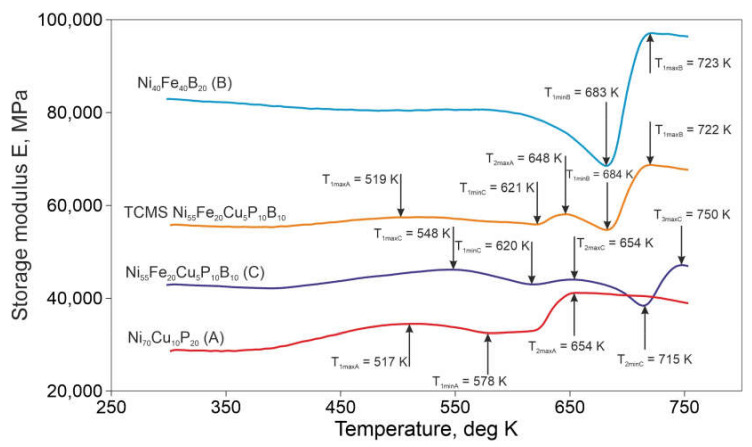
Storage modulus E’ versus temperature plots derived from DMTA (dynamic mechanical thermal analysis). The following symbols: T_1maxA_, T_1minA_, T_2maxA_, T_1minB_, T_2maxB_, T_1maxC_, T_1minC_, T_2maxC_, T_2minC_, and T_3maxC_ denote successive maxima and minima of the storage modulus E’ occurring in alloys A, B, and C, respectively, and occurring in the TCMS alloy and assigned to the structural components corresponding to alloys A, B, and C.

**Table 1 materials-14-01741-t001:** Phases present in the TCMS Ni_55_Fe_20_Cu_5_P_10_B_10_ alloy during in situ X-ray diffraction measurements at different temperatures.

Temperature (K)	Phases					
	Ni	Ni_3_P	CuNi	FeNi	FeNi_3_	Fe_1.5_Ni_1.5_B
293	-	-	-	-	-	-
553	-	-	-	-	-	-
603	●	●	●	-	-	-
623	●	●	●	-	-	-
643	●	●	●	-	-	-
663	●	●	●	-	-	-
713	●	●	●	●	●	●
753	●	●	●	●	●	●
803	●	●	●	●	●	●

**Table 2 materials-14-01741-t002:** The values of peak temperatures for different heating rates in Ni_55_Fe_20_Cu_5_P_10_B_10_ (**C**), TCMS Ni_55_Fe_20_Cu_5_P_10_B_10_, Ni_40_Fe_40_B_20_ (**B**), Ni_70_Cu_10_P_20_ (**A**) alloys.

*β*, K/min	T_1A_	T_1C_	T_2A_	T_1B_	T_2C_	T_3A_
Ni_55_Fe_20_Cu_5_P_10_B_10_ (C)						
10	-	634	-	-	727	-
20	-	642	-	-	733	-
40	-	652	-	-	739	-
60	-	658	-	-	744	-
80	-	671	-	-	753	-
TCMS Ni_55_Fe_20_Cu_5_P_10_B_10_						
10	588	-	642	711	728	-
20	598	-	651	717	735	-
40	609	-	662	725	744	-
60	617	-	668	730	746	-
80	621	-	674	735	750	-
Ni_40_Fe_40_B_20_ (B)						
10	-	-	-	706	-	-
20	-	-	-	712	-	-
40	-	-	-	718	-	-
60	-	-	-	724	-	-
80	-	-	-	727	-	-
Ni_70_Cu_10_P_20_ (A)						
10	591	-	641	-	-	722
20	598	-	649	-	-	734
40	606	-	658	-	-	747
60	613	-	663	-	-	754
80	618	-	668	-	-	760

**Table 3 materials-14-01741-t003:** The values of energy activation [kJ/mol] derived from the Kissinger method for peak temperatures for different heating rates in Ni_55_Fe_20_Cu_5_P_10_B_10_ (**C**), TCMS Ni_55_Fe_20_Cu_5_P_10_B_10_, Ni_40_Fe_40_B_20_ (**B**), Ni_70_Cu_10_P_20_ (**A**) alloys.

E_1A_	E_1C_	E_2A_	E_1B_	E_2C_	E_3A_
Ni_55_Fe_20_Cu_5_P_10_B_10_ (C)					
-	192	-	-	373	-
TCMS Ni_55_Fe_20_Cu_5_P_10_B_10_					
178	-	224	365	415	-
Ni_40_Fe_40_B_20_ (B)					
-	-	-	406	-	-
Ni_70_Cu_10_P_20_ (A)					
223	-	264	-	-	243

## Data Availability

The data presented in this study are available on request from the corresponding author.

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
