# Peer review of "Microstructure Development and Properties of the Two-Component Melt-Spun Ni55Fe20Cu5P10B10 Alloy at Elevated Temperatures"

_materials, 2021, doi:10.3390/ma14071741_

Round 1

Reviewer 1 Report

The authors investigated the features, microstructure, phase composition, mechanical properties, and thermal stability of a two-component melt-spun Ni-Fe-Cu-P-B alloy. It was reported that there was a band-like microstructure in the composite alloy that remained stable up to 800 K. Furthermore, the authors mentioned that the storage modulus was dependent of the modulus changes occurring in the amorphous alloys that are components of the composite. Interesting article, but needs further work.

1. Some of the figures are hard to read. For instance, the text on Fig. 1 is too small and must be increased in size.

2. For the nanoindentation tests, why was the recorded hardness and modulus based on one indent? It is strongly recommended that the authors obtain the above results based on ten indents for statistical certainty.

3. Please elaborate more on your experimental work. For instance, what was the X-ray energy for the X-ray diffraction experiments. Also, how were the crystalline phases determined?

4. The authors studied the crystallization behavior of metallic glasses with varying composition. However, there was little to no discussion as to the role of chemical complexity on the crystallization of the glass. The authors should therefore give a more thorough discussion on this subject. The references below should help:

Journal of Applied Physics 108(10) (2010) 8, Intermetallics 116 (2020) 106655, Acta Phys. Sin. 66(17) (2017) 9.

5. In the conclusions section, please better explain how your work resulted in an advance.

Author Response

List of changes

We are very grateful to the reviewers for the time devoted to reading the work carefully and for comments on our article submitted to your esteemed journal.

We hereby declare that we agree with all comments regarding our work. The article has been revised taking into account all comments submitted in both reviews. We hope that the current quality of this work meets these requirements and that it is suitable for publication in your journal.

See attached file for the changes in our submission.

Reviewer 2 Report

Abstract - generally good, some English language issues in the last 2 sentences. Missing broader implications of the results.

Introduction - very short, many references for single sentences are not explored and compared. TCMS is not explained in sufficient detail. Significant expansion required.

Odd change in font line spacing for 64?

Please split the text into paragraphs for ease of reading.

Experimental - many key parameters missing. There should be sufficient information as to allow an independent researcher to repeat the study e.g. SEM detectors, voltage, current? Please expand.

Fig 1 - far too much detail on a single plot - font too small to read, generally ineffective - suggest overlaying plots? Please put it next to where it is first referred to.

Generally - put figures close to first mention.

The comparisons in Figure 2 are unclear on such magnified plots - are these images representative of the entire surface?

Fig 3 - suggest using colours to highlight different lines. Odd theta?

Fig 4 - small fonts, difficult to read - maybe overlay indentation results? Lots of images - all needed? 

The layout of the paper is really difficult to follow with papers far from their location. It also seems as though the same information has been presented multiple times.

Figure 6 is stretched and has very small text. Please consider how to improve.

Figure 7 - busy with lots of detail - consider the use of colour.

Lack of paragraphs make the text difficult to read.

Overall this article is very poorly organised and presented. There is a significant amount of data presented in difficult to understand formats. The lack of structure makes the text difficult to read and understand. It is possible that this article contains useful information but at present I cannot follow the discussion or results due to the overwhelming avalanche of information. I suggest that the authors carefully review the information that is key to what they are trying to convey (do not just include everything), restructure and refine the text and layout, improve the figures and resubmit.

Author Response

List of changes

We are very grateful to the reviewers for the time devoted to reading the work carefully and for comments on our article submitted to your esteemed journal.

We hereby declare that we agree with all comments regarding our work. The article has been revised taking into account all comments submitted in both reviews. We hope that the current quality of this work meets these requirements and that it is suitable for publication in your journal.

Below we present changes in this work, taking into account the comments of reviewers:

Reviewer #2:

Abstract - generally good, some English language issues in the last 2 sentences. Missing broader implications of the results.

The two last sentences of the abstract were removed and re-edited, thus supplementing the broader implications of the results. These sentences are as follows: “9.   This work shows the unique properties of the TCMS composite in comparision with the melt-spun alloy of the same nominal chemical composition that was produced traditionally. The mechanical and the thermal properties of this composite are obviously inherited from the amorphous state of the alloys that are precursors for the two-component melt spinning.”

Introduction - very short, many references for single sentences are not explored and compared. TCMS is not explained in sufficient detail. Significant expansion required.

Introduction was re-edited and the meaning of its sentences is now clarified. Furthermore TCMS is explained and introduced in the Experimental section. A diagram showing this method of remelting and the location of the partition separating the two alloys during the ejection onto a rotating copper roller was introduced in the text as Figure 1.

Odd change in font line spacing for 64?

The odd change in font line spacing comes possibly fro the template of the Journal. We are not changing this with hope that this is not a substantive problem.

Please split the text into paragraphs for ease of reading.

The text was corrected and splited into paragraps for ease of reading.

Experimental - many key parameters missing. There should be sufficient information as to allow an independent researcher to repeat the study e.g. SEM detectors, voltage, current? Please expand.

This corrected section of the work is as follows: “The cross-sections of the ribbons were polished mechanically and etched with a pure nitric acid. The metallographic specimens intended for testing on the nanoindenter were not etched. Then the morphology  and  chemical  composition  of  the  cross-section of these TCMS heat treated samples as well as the samples in as-melt-spun state were analysed by scanning electron microscope (SEM) JEOL 6610 with an Oxford X-Ray micro-analyzer. The observations were made in the secondary electron image mode under accelaration voltage 20kV and working distance 10mm.”

Fig 1 - far too much detail on a single plot - font too small to read, generally ineffective - suggest overlaying plots? Please put it next to where it is first referred to.

Figure 1 is after the corrections Figure 2. This Figure is put next to where it is first referred to. It is also re-edited in order to improve the readibility, and the larger font is also applied.

Generally - put figures close to first mention.

All Figures in the text are put close to their first mention.

The comparisons in Figure 2 are unclear on such magnified plots - are these images representative of the entire surface?

Figure 2 is now labeled as Figure 3. The magnifications were unified for feasibility of comparisons between the variants. The images included in this Figure are representative for whole section of the ribbon. The picture magnification was as high as it was possible - it is only few mm smaller than the thickness of the ribbon.

Fig 3 - suggest using colours to highlight different lines. Odd theta?

Figure 3 now labeled as Figure 4 is corrected. The colours were applied in order to highlights different lines. The font uset for theta was corrected. 

Fig 4 - small fonts, difficult to read - maybe overlay indentation results? Lots of images - all needed?

Figure 4 now labeled as Fig 5 is corrected. As suggested, the indentation results were overlayed and some redundant curves were eliminated from the picture.

The layout of the paper is really difficult to follow with papers far from their location. It also seems as though the same information has been presented multiple times.

The submission was scanned for passages where the same information is repeated and the location of the information is correlated with the corresponding Figures and references.

Figure 6 is stretched and has very small text. Please consider how to improve.

Figure 6 now labeled as Figure 7 was corrected. Now this Figure is re-edited in order to make it more clear. The font sizes were also increased.

Figure 7 - busy with lots of detail - consider the use of colour.

Figure 7 now labeled as Fig 8 was corrected. Unnecessary details such as equations were removed. The colour was used in order to improve readibility.

Lack of paragraphs make the text difficult to read.

The text is now divided into paragraphs, which should make it eaisier to read.

Overall this article is very poorly organised and presented. There is a significant amount of data presented in difficult to understand formats. The lack of structure makes the text difficult to read and understand. It is possible that this article contains useful information but at present I cannot follow the discussion or results due to the overwhelming avalanche of information. I suggest that the authors carefully review the information that is key to what they are trying to convey (do not just include everything), restructure and refine the text and layout, improve the figures and resubmit.

Organization and ways of presentation were corrected in order to improve the readibility of the submission. The formats of the presented data were corrected and made easy to read. The structure of the paper was corrected. The authors carefully reviewed the paper in order to highlight and expose the crucial results and information. The text and layout were improved. – Thus we submit the paper with major changes. The some uncecessary sections of text have been crossed out and replaced with corrected text marked “red.

Round 2

Reviewer 2 Report

Generally the manuscript is much improved. Minor corrections to tables and Fig 1 remaining that can be achieved during the proofing process.

Figure 1 - can you not put an image? Border not required.

Table 2 + 3 - very sparse - there must be a better way of presenting this data